# Essential Factors Enhancing Industrialized Building Implementation in Malaysian Residential Projects

**Al-Hussein M. H. Al-Aidrous** [1,*], **Nasir Shafiq** [1,*], **Yasser Yahya Al-Ashmori** [1], **Al-Baraa Abdulrahman Al-Mekhlafi** [2] and **Abdullah O. Baarimah** [1]

1   Department of Civil & Environmental Engineering, Universiti Teknologi PETRONAS, Seri Iskandar 32610, Perak, Malaysia
2   Department of Management & Humanities, Universiti Teknologi PETRONAS, Seri Iskandar 32610, Perak, Malaysia
*   Correspondence: hus.alaidrous@gmail.com (A.-H.M.H.A.-A.); nasirshafiq@utp.edu.my (N.S.)

**Abstract:** Sustainable residential development requires a balance between the increasing demand for housing and the efficient use of materials and resources. The increasing use of industrialized building systems (IBSs) through new building techniques and materials holds high potential as an optimum construction alternative. Although considerable research has been conducted on industrialized buildings, very few studies have focused on low- and mid-rise residential buildings. Therefore, this paper aims to fill this gap. An extensive literature review was conducted to identify the critical success factors (CSFs) followed by an interview to discuss and validate the collected factors. This study resulted in twenty-six factors grouped into five CSFs comprising planning and control, roles and responsibilities, policies and incentives, industry maturity and technology advancement. In addition, 219 survey responses were collected and analyzed. Three factors were perceived differently among organizations including commitment toward IBS policy, implementation of preferential policy for IBSs and imposition of higher taxes on waste dumping. The top five CSFs were early planning to implement IBSs, extended training for local labor, effective communication among project players, project location evaluation and accessibility and standardized design concept adoption. The findings of this paper will help policymakers to review current practices and help develop a roadmap for sustainable IBS development for all industry organizations.

**Keywords:** prefabricated buildings; critical success factors; offsite construction (OSC); enablers; sustainable building

## 1. Introduction

Global attempts to reform the construction sector to achieve better utilization of modern methods of construction were spurred by the need to tackle the acute housing shortage hurdle [1]. Increasing demand for residential buildings has led to a housing crisis worldwide, especially in developing countries [2]. Moreover, the building sector is a major waste producer and consumer of materials and natural resources [3,4]. The construction industry is faced with growing needs for residential buildings. However, in the same way, it is required to reduce the use of natural resources with minimum environmental impact. In response to that, several countries are being pushed to take new measures to boost the building rate without deteriorating the environment [5,6]. Industrialized construction is considered a sustainable and efficient construction method providing a better response to the housing market supply and demand dynamics in a sustainable manner [7].

The efficient supply of affordable housing is a persistent challenge. Limited supply and so much demand are usually the norms in the housing market, even in developed countries such as Australia [1]. In Malaysia, from the first housing plan in the middle of the last century (1956) until the most recent plan of the National Housing Policy (2018–2025),

the housing issue still persists [8,9]. The latest plan announced by the previous government in 2018 was intended to supply one million affordable housing units in the upcoming ten years. However, the following government has suspended the plan since there are around 20% of high-rise affordable apartments that are overhung. Thus, integrated planning is needed to effectively know the appropriate location and housing type needs to be delivered, indicating the importance of low-mid-rise buildings [10–12].

Although industrialized buildings are not new, few countries were able to expand their use successfully. Different terminologies are used in other countries, including offsite construction (OSC), modular construction and prefabricated buildings. Advanced countries such as Japan, Singapore, several European countries, the USA, Canada and more recently China were able to shift their building practices toward more industrialized methods [13]. On the contrary, countries like Malaysia, with low manufacturing capabilities, have difficulty moving away from conventional construction, mainly when the industry is made up of a high volume of small and medium construction companies [14]. In Europe, around 8000 manufacturing plants are producing prefabricated/precast concrete components. These plants contribute 25% of the total concrete consumption in the European industry [15]. Since 2016, Chinese authorities have issued various policies and incentives to support and increase industrialized buildings' position [16]. As a result, in only three years, industrialized buildings account for over 13% of newly developed buildings in China [17]. The Chinese example gives a clear indication that applying industrialized buildings would flourish with strong support and commitment from the government and the industry.

Despite the successful implementation of industrialized buildings in developed nations, their application in developing countries is still lagging [18]. Wuni and Shin [19] noted that one of the challenges for successfully managing construction works is identifying the pertinent critical success factors (CSFs) in different project types and territories. Considerable research efforts have been conducted on critical success factors in general [19–21], yet very few have focused on low-mid-rise buildings. Therefore, these studies could not reflect recent developments in the construction of residential projects since low-rise buildings are the prevailing ones in many different countries. Based on previous studies, Lin et al. [22] explored the applicability potential of IBSs in Australian low-rise buildings, whereby MacAskill et al. [1] investigated the supply chain strategies for small and medium-sized apartments in Australia as well. Additionally, Brissi et al. [23] examined the use of industrialized buildings in low-rise multifamily housing projects in the United States. Even though these studies investigated important aspects of low-rise buildings in developed countries, there is a lack of similar studies in developing countries. Therefore, there is a need to analyze the critical success factors driving the adoption of IBSs in low- and mid-rise buildings in developing countries such as Malaysia. Hence, this study was conducted to fill this gap. Moreover, this research utilized a mixed-method approach to investigate the CSFs in Malaysia. The reason for using the mixed method was to first investigate the critical factors from the literature based on Malaysian experts through an interview and then measure the finalized data from the interview on a wider number of professionals using the questionnaire.

This research offers practical and theoretical contributions to IBS implementation. Theoretically, this study expands the body of knowledge regarding critical factors that contribute to enhancing industrialized buildings' development. Likewise, this study provided a list of CSFs for low-rise buildings which may be considered a reference point for future studies in different contexts and project types. Practically, this research can be viewed as a direction for industrialized building project managers to help them assess the essential factors in ranked order which measure the success of IBS projects and provide guidance for IBS stakeholders. The remaining of this paper is as follows. First, Malaysian building sectors are discussed and analyzed. Second, methodology and interview outcomes are presented. Third, results, data analysis and discussion are followed. Finally, the conclusion, limitations and suggestions for future studies are presented.

## 2. The Malaysian Building Sector

In developed countries, industrialized building technologies continue to show great opportunities in terms of housing supply. Yet, building projects in many developing countries continue to face difficulties, especially with the rising urbanization and population expansion, both of which worsen the scarcity of housing projects [24]. In Malaysia, the demand for affordable housing has constantly been a pressing issue [25]. The participation of IBSs in the residential sector is limited regardless of the government-mandated adoption of IBSs for large public projects [26]. Ebekozien et al. [27] suggested adopting IBSs as a key approach to reduce building costs and boost the housing supply in Malaysia. Countries such as Sweden, China and Australia have increased the delivery of housing projects using industrialized construction [22,28,29]. Thus, the large volume required for current and future housing projects will not be achieved unless residential building construction shifts to manufacturing production practices.

In 2021, the total property units in Malaysia reached 6.9 million units, diversified from residential and semi-residential units to commercial buildings such as shops and industrial buildings [30]. In the total property stock, residential buildings represent the highest percentage of 85.6%, while the remaining include: shops (7.8%), serviced apartments (4.1%), SOHO (acronym for Small Office, Home Office) (0.7%) and industrial (1.7%) as shown in Table 1.

**Table 1.** Total Property Units.

| Property Type | Property Number | Percentage | Purpose of Use | Surfaces |
|---|---|---|---|---|
| Residential | 5933,254 | 85.62% | Residential use only |  |
| Shops | 542,063 | 7.82% | Commercial use only |  |
| Serviced Apartment | 287,022 | 4.14% | Residential building commercially titled |  |
| SOHO | 48,607 | 0.70% | Mix of offices and residential areas, acronym for Small Office, Home Office | |
| Industrial | 118,708 | 1.71% | Building or structures used for manufacturing and storing |  |
| Total | 6929,654 | 100% | | |

The residential market comprises landed housing, low-rise, mid-rise and high-rise buildings. As shown in Figure 1, the landed housing includes terrace (41%), semi-detached (7%), detached (8%), townhouse (1%), cluster (1%) and low-cost houses (12%). Additionally, low- and mid-rise buildings such as flats account for 14%, and apartments and condomini-

ums represent 16%, as medium-to-high-rise buildings. Landed housing dominates the market supply with 69.9% of all residential buildings.

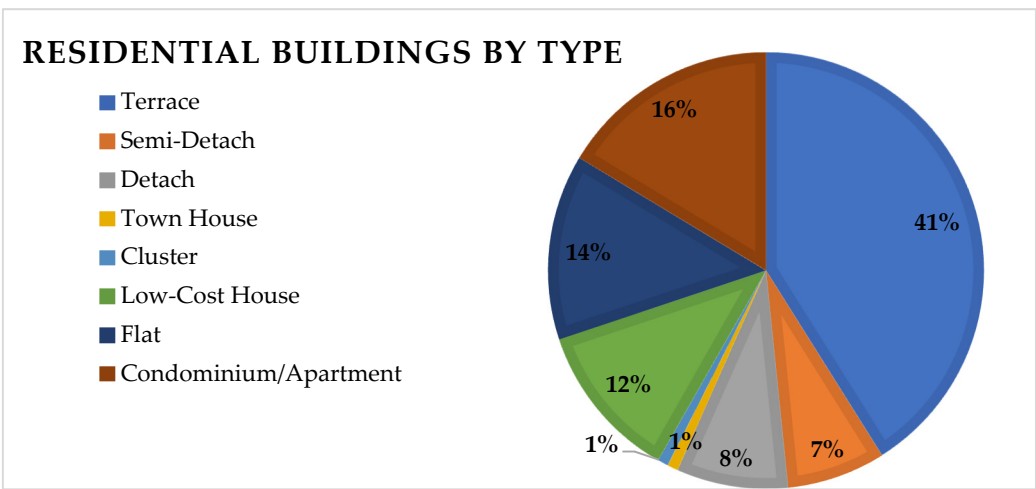

**Figure 1.** Types of residential buildings.

Furthermore, landed housing, low-rise and mid-rise buildings represent a total market share above 83%, as indicated in Table 2. Since this building segment represents the vast majority of the building market, it requires more investigation to push the industrialization effort to a larger scale. Due to the high impact of this market sector and since very little research has investigated low-mid-rise buildings, this research focused on this market segment and paving the way for successful IBS implementation.

**Table 2.** Low-, mid- and high-rise residential buildings.

| Type of Buildings | Number of Supplied Units | Percentage | Accumulative | Specific Type | Height | Stories | Surfaces |
|---|---|---|---|---|---|---|---|
| Landed houses | 4146,771 | 69.9% | 69.9% | Terrace, Semi-Detached, Detached, Townhouse, Cluster and Low-Cost House | 10 m | 1–2 Stories | |
| Low and mid-rise buildings | 815,183 | 13.7% | 83.6% | Flat and low-cost flat | 11–35 m and 36–69 m | 3–5 Stories and 6–16 Stories | |
| High-rise buildings | 971,300 | 16.4% | 100% | Condominium/ Apartment | 70 m | Above 16 Stories | |
| Total | 5933,254 | | | | | | |

## 3. Methods

This research used a mixed methodology of qualitative and quantitative approaches including a thorough review of the literature, semi-structured interviews and survey questionnaires. This process allows triangulation of both approaches [31]. In the construction management domain, various researchers acknowledged the importance of using a mixed-method approach [32,33]. The first and foremost step is to learn about a specific matter

in terms of its valuable results as well as the methodological aspects [32,34,35]. Thus, this research thoroughly reviewed the literature to determine the critical success factors of industrialized buildings. Moreover, to refine and ensure the suitability of the data collected from the literature, conducting interviews with industry experts is a viable and proven strategy [36,37]. Interviewing industry experts in Malaysian construction is crucial to discuss, refine and validate the factors collected from the literature and verify their relatedness to the Malaysian local industry. The interview established a set of main and sub-factors of CSFs. This process enriched this study as it provided valuable inputs from the industry in practice to be involved in conducting the questionnaire. Furthermore, a survey questionnaire was developed to identify and rank the importance of the selected factors from the interview phase. Then, several statistical analysis tests were performed including the normality test, Kruskal–Wallis test and ranking analysis. A detailed flow of the research method is shown in Figure 2.

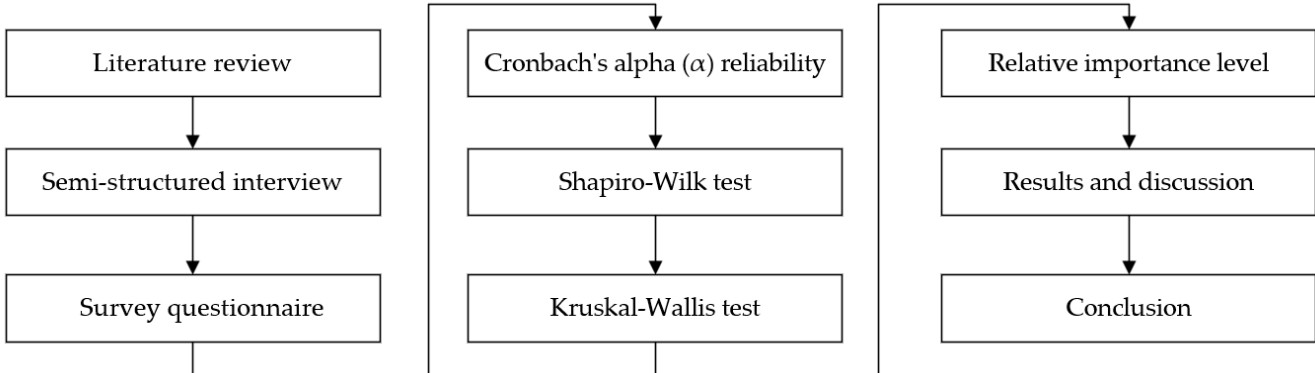

**Figure 2.** Research method workflow.

## 3.1. Interview

The interview is a data collection tool that can collect valuable and detailed information about the studied matter. This technique can collect up-to-date information and experiences [38]. This study conducted interviews to reveal the Malaysian experts' perspectives on the industrialized buildings sector. Thirty potential experts from the industry and academia were contacted for an interview. An email was sent to all individuals explaining the aim and procedure of the interview to shorten the time needed to explain the interview's purpose. Only fifteen experts were available and agreed to attend the interview. The experts interviewed ranged from manufacturers, government officials and design consultants to academicians. The findings of the interviews added additional success factors to the ones found in the literature worldwide, leading to an up-to-date investigation. Therefore, this process ensured the applicability of the literature review collected to the recent developments in the local Malaysian industrialized construction.

Table S1 shows the profile of the professionals interviewed. The majority of interviewees held experience of more than ten years. Most of them had positions as executives and design managers, holding two vital administrative jobs. Additionally, the wide range of respondents including developers, manufacturers, consultants and government officials provided a comprehensive perspective.

The interview enables thorough discussion through open-ended questions based on the research aim. The primary goal of these interviews was to review and compile a list of critical success factors (CSFs) that are relevant to the Malaysian residential building sector. Fifteen semi-structured interviews were undertaken with IBS professionals in order to evaluate the prepared list of CSFs and propose any additional factors, which then would be added to the survey questionnaire. The main output targeted by the interviews included the interviewees' background information, the pace of progress in low-rise buildings and key success factors for successful IBS implementation. Moreover, the experts were asked

whether the compiled factors from the literature would be considered success factors for IBS projects in Malaysia and if any other additional factors can be added.

Additionally, interviewees were asked about each factor's group and the appropriate name for each group. The interviews yielded a list of twenty-six CSFs divided into five groups deemed appropriate for the Malaysian building sector. A number of factors were added based on the experts' perspectives. The final list of factors is shown in Table 3.

**Table 3.** The final list of critical success factors.

| Category | No. | Critical Success Factors | Sources |
| --- | --- | --- | --- |
| Policies and Incentives | 1 | Clear plan and policy that can ensure IBS implementation. | [20,39] |
| | 2 | Commitment of agencies and local authorities in the state toward the implementation of IBS policy. | Interview |
| | 3 | Implementation of preferential policy for IBS which can motivate developers and buyers to adopt IBS. | [39,40] |
| | 4 | Adoption of non-financial incentives for IBS, e.g., faster approval procedure, exemption from some building requirements. | [41] |
| | 5 | Providing incentives for IBS implementation including tax privilege and loan support. | [39,42] |
| | 6 | Implementing higher taxes and penalties for building waste dumping. | [16,42,43] |
| Roles and Responsibilities | 7 | Adopting role and business strategies that support IBS. | [14,29,44] |
| | 8 | Manufacturer involvement role in design and construction. | [45,46] |
| | 9 | Manufacturer readiness to provide training before installation of components. | Interview |
| | 10 | Adoption of procurement system that suits IBS construction method. | [47–49] |
| | 11 | Implementation of standard procedure for onsite and offsite inspection/supervision work. | Interview |
| Planning and Control | 12 | Early planning to implement IBS system. | [19,45,50] |
| | 13 | Team agreement on project deliverables. | [50,51] |
| | 14 | Adoption of standard dimensions and modular coordination to reduce cost. | [44,45,52] |
| | 15 | Freezing the design early to reduce any possible rework. | [19,50] |
| | 16 | Effective communication and collaboration among players from early phase. | [19,45] |
| | 17 | Project location evaluation and accessibility. | [45,53] |
| Industry Maturity | 18 | Increasing the number of high-quality IBS housing units will increase buyers' acceptability. | [26,45,46] |
| | 19 | Sufficient experience of contractors and designers in IBS. | [45,54,55] |
| | 20 | Skills and competency of project players. | [20,50,56] |
| | 21 | Competitive labor wage rate. | [50,57] |
| | 22 | Extended training for local labor to strengthen skills in IBS. | Interview |
| Technology Advancement | 23 | Ensuring effective design and installation using building information modeling (BIM). | [58–60] |
| | 24 | Continuous R&D to improve current practices and reach competitive advancement. | Interview |
| | 25 | Using at least a semi-automated production toward automation and robotic construction. | [19,39] |
| | 26 | Adopting advanced technologies including cloud and real-time collaboration, advanced building materials and internet of things (IoT). | [19,61] |

*3.2. Questionnaire*

In the construction research field, a questionnaire survey is a popular method for gathering the views of industry professionals on a certain issue [62–64]. As a result, a questionnaire survey was employed to gauge professionals' perspectives on the collected critical success factors. Following an extensive literature review and semi-structured interviews, a survey questionnaire was established. The two steps prior to survey establishment addressed and mitigated any potential threat to the survey content validity. The feedback from the interviews was used to improve the questionnaire and provide additional factors for the current Malaysian IBS industry. The first part aimed to obtain the respondents' background, such as years of experience, involvement level in industrialized buildings, business nature, job designation and company size. In the second part, respondents were asked to rate the critical success factors based on a five-point Likert scale (1: highly insignificant, 2: insignificant, 3: neutral, 4: significant, 5: highly significant). In the third part

of the questionnaire, an open-ended question asked the respondents to add any further comments on the use of industrialized construction in residential buildings. The 5-point rating scale was used to compare different organization types and rank the factors due to their mean and relative importance.

## 4. Results and Data Analysis

### 4.1. Background Information

The data and responses obtained from the questionnaire were analyzed through statistical methods. A number of statistical analyses were performed supported by the sufficient number of responses collected. The first of these analyses was to evaluate the collected data and measure its reliability and internal consistency by computing Cronbach's alpha ($\alpha$) coefficient. Secondly, the Shapiro–Wilk test was used to check the normality of the dataset. Based on the output of the normality test, the Kruskal–Wallis test was conducted which is used to measure the agreement among different groups of respondents. Lastly, the relative importance index of the critical success factors was measured and reported in a ranked order. This study targeted the companies registered in the government database as industrialized building companies including manufacturers, contractors and consultants [65]. The database contains the company name, address and email. An email questionnaire was sent to these potential companies afterward with one or two reminders. Eventually, a total of 219 responses were collected. However, 16 responses indicated an experience in only high-rise buildings and thus were excluded. Finally, a total of 203 responses were deemed sufficient and related to low-mid-rise buildings. Based on Figure 3, 70.3% were involved in landed houses, 61.9% in low-rise buildings, 32.2% in medium-rise buildings and 19.3% in high-rise buildings. It must be noted that each respondent was given a choice to select their involvement in one or more building types. Respondents with only experience in high-rise buildings were ignored, while respondents who held experience in both low-rise and high-rise buildings were considered valid for further analysis. It is also clear that most respondents were experienced in low-rise buildings as shown in Figure 3.

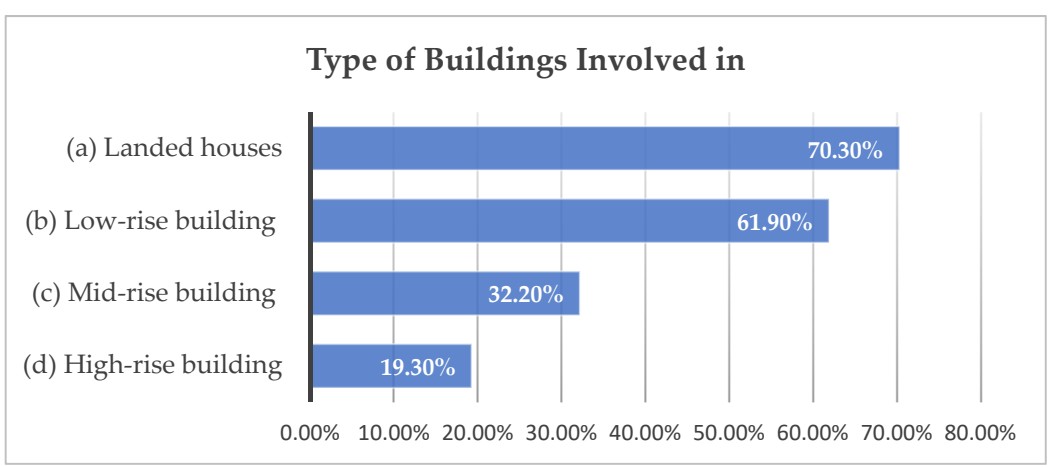

**Figure 3.** Respondents' involvement in different building types.

Table 4 shows respondents' profiles and backgrounds. The respondents were from various organizations including contractors, consultants, IBS manufacturers, government officials and developers. This diversity in the collected responses represents the whole range of the industry professionals and increases data reliability leading to reliable results. Contractors represent the largest group with 50% of the total respondents since the industry consists of a large number of small and medium construction companies. In addition, the positions held by respondents ranged from the director, senior manager, manager, design engineer and site engineer to quantity surveyor.

**Table 4.** Profiles of questionnaire respondents.

| Qualification | Sub-Group | Responses | % Responses |
|---|---|---|---|
| Education level | High School | 16 | 7.9 |
| | Diploma | 45 | 22.2 |
| | Degree | 85 | 41.9 |
| | Master | 47 | 23.2 |
| | Ph.D. | 10 | 4.9 |
| Organization type | Contractor | 103 | 50.7 |
| | Consultant | 53 | 26.1 |
| | Government | 22 | 10.8 |
| | IBS manufacturer and contractor | 7 | 3.4 |
| | IBS manufacturer | 12 | 5.9 |
| | Developer/Client | 6 | 3.0 |
| Job role | Director | 52 | 25.6 |
| | Senior manager | 13 | 6.4 |
| | Manager | 67 | 33.0 |
| | Design engineer | 28 | 13.8 |
| | Site engineer/quantity surveyor | 43 | 21.2 |
| Construction experience | Less than 5 years | 38 | 18.7 |
| | 5 to 10 years | 53 | 26.1 |
| | 10 to 15 years | 34 | 16.7 |
| | More than 15 years | 78 | 38.4 |
| Experience in IBS | None | 57 | 28.1 |
| | Less than 5 projects | 97 | 47.8 |
| | 5 to 10 projects | 21 | 10.3 |
| | 11 to 15 projects | 11 | 5.4 |
| | More than 15 projects | 17 | 8.4 |

Regarding job roles, directors and managers represented 25% and 33% of all respondents. Additionally, over 80% of respondents had experience of more than five years, and 55% held experience of over ten years. Nevertheless, only 24% were experienced in more than five IBS projects while the majority (around 50%) had experience in less than five IBS projects. This outcome is justifiable as IBSs are at an early stage of implementation worldwide, whereby their application in Malaysia is limited to large public projects. It is worth noting that, even though all collected responses were for IBS companies that are listed in the governmental directory [65], more than a quarter of respondents held no real experience in IBSs, showing that the Malaysian construction industry is still at the infancy level of implementing industrialized building practices.

*4.2. Reliability Test and Agreement among Respondents*

The first statistical analysis this study conducted in a series of analyses, with the aid of the Statistical Package of Social Science (SPSS), was to evaluate the collected data and measure their reliability and internal consistency through calculating Cronbach's alpha ($\alpha$) coefficient. According to Tavakol and Dennick [66], the minimum value for the alpha coefficient is 0.70. In this study, the result of the reliability analysis of all twenty-six of CSFs yielded a value of 0.94. Since the coefficient of the factors was above the threshold of 0.70, the result reflects a high level of internal consistency among respondents indicating valid and reliable questionnaire data for further analysis.

Two key comparison techniques were used to evaluate the agreement among the responses of different organization types such as contractors, manufacturers and consultants, including one-way ANOVA and the Kruskal–Wallis test. If the data are normally distributed, the parametric one-way ANOVA test should be used for multi-group comparison. Otherwise, the non-parametric Kruskal–Wallis test does not require a normal distribution assumption [67]. To measure data normality, the Shapiro–Wilk test was used. In other words, if the *p*-value yielded from the normality test is more than 0.05 at a confidence

level of 95%, the data are normally distributed, and a parametric test should be employed. However, according to Table 5, Shapiro–Wilk test outcomes in this study generated a value of less than 0.05 for all factors, indicating that the data are not normal. Thus, the non-parametric test was utilized. The Kruskal–Wallis test is a non-parametric test used to evaluate if there are any statistically significant variations in professionals' responses within the groups investigated. As mentioned by Field [67], there are significant differences among different groups of respondents when the *p*-value generated from the Kruskal–Wallis test is lower than 0.05 at a 95% confidence level. Table 5 shows that there is no significant difference among different organizations except for three factors, CSF2 "Commitment of agencies and local authorities in the state toward the implementation of IBS policy", CSF3 "Implementation of preferential policy for IBS which can motivate developers and buyers to adopt IBS" and CSF6 "Implement higher taxes and penalties for building waste dumping".

*4.3. Relative Importance Level of Critical Success Factors*

This section aims to determine the importance level of various CSFs for IBS implementation. This is the second step after evaluating the data reliability and the agreement among respondents' groups. Two methods were applied to rank the CSFs including the mean score and standard deviation as well as the index of relative importance (IRI). The assessment of CSF importance was based upon a five-point rating scale. For each factor measured, there is a core value called the mean. The calculation process of the mean is based on multiplying all the individual scores by the frequency divided by the total replies. Subsequently, the standard deviation is calculated to measure the variation within the collected dataset. Drawing on Saad et al. [21] work, this study calculated the index of relative importance (IRI). The highest possible value for the IRI is 1; thus, a higher IRI value indicates higher importance of the factor. Generally, the IRI value varies from 0.2 to 1, whereby the IRI would reach a 0.2 value if all respondents selected (1) extremely disagreed, and the IRI would equal 1 if all respondents selected (5) extremely agreed. The values are categorized into four groups: above 0.8 is very high, above 0.6 high, above 0.4 low and above 0.2 very low [68]. The equation for calculating the IRI is shown below:

$$Index\ of\ Relative\ Importance\ (\text{IRI}) = \frac{\sum W_i}{(A*N)}$$
$$Index\ of\ Relative\ Importance\ (\text{IRI}) = \frac{1*n_1+2*n_2+3*n_3+4*n_4+5*n_5}{(5*N)} = (1) \tag{1}$$

where $W_i$ is the weight of each factor which ranges from 1 to 5, *A* represents the highest weight which in this study is 5, as this study is using a five-point rating scale, and finally, *N* is the total collected responses which in this study is 219. Table 6 shows the mean, standard deviation, IRI, ranking in each group and the overall ranking. The top five CSFs are: (1) early planning to implement IBS system, (2) extended training for local labor to strengthen skills in IBS, (3) effective communication and collaboration among players from the early phase, (4) project location evaluation and accessibility and (5) adoption of standard dimensions and modular coordination to reduce cost.

The mean and standard deviation are demonstrated in Table 6. The factor with the highest mean is early planning to implement the IBS system (mean CSF12: 4.468). In contrast, the lowest mean is implementing higher taxes and penalties for building waste dumping to reduce dependence on conventional construction (mean CSF6: 3.956). Additionally, the factor with the lowest standard deviation (SD) is extended training for local labor to strengthen skills in IBS (SD CSF22: 0.711), which is ranked second in the overall ranking whereby the factor with the highest standard deviation is implementing higher taxes and penalties for building waste dumping to reduce dependence on conventional construction (SD CSF6: 1.078).

**Table 5.** Rating frequency, Shapiro–Wilk normality and Kruskal–Wallis Tests.

| Code | Critical Success Factor | Rating Frequency | | | | | Shapiro–Wilk * | Kruskal–Wallis | |
|---|---|---|---|---|---|---|---|---|---|
| | | 1 | 2 | 3 | 4 | 5 | | Chi Square | Sig. * |
| CSF1 | Clear plan and policy that can ensure IBS implementation. | 0 | 2 | 44 | 72 | 85 | 0.000 | 6.910 | 0.227 |
| **CSF2** | Commitment of agencies and local authorities in the state toward the implementation of IBS policy. | 1 | 5 | 44 | 71 | 82 | 0.000 | 14.517 | 0.013 * |
| **CSF3** | Implementation of preferential policy for IBS which can motivate developers and buyers to adopt IBS. | 0 | 3 | 35 | 76 | 89 | 0.000 | 16.428 | 0.006 * |
| CSF4 | Adoption of non-financial incentives for IBS, e.g., faster approval procedure, exemption from some building requirements. | 1 | 6 | 37 | 79 | 80 | 0.000 | 9.405 | 0.094 |
| CSF5 | Providing incentives for IBS implementation including tax privilege and loan support. | 2 | 2 | 32 | 62 | 105 | 0.000 | 7.048 | 0.217 |
| **CSF6** | Implementation of higher taxes and penalties for building waste dumping. | 8 | 7 | 47 | 61 | 80 | 0.000 | 13.075 | 0.023 * |
| CSF7 | Adopting role and business strategies that support IBS. | 0 | 2 | 31 | 90 | 80 | 0.000 | 4.803 | 0.440 |
| CSF8 | Manufacturer involvement role in design and construction. | 2 | 7 | 32 | 78 | 84 | 0.000 | 7.637 | 0.177 |
| CSF9 | Manufacturer readiness to provide training before installation of components. | 0 | 2 | 26 | 70 | 105 | 0.000 | 1.232 | 0.942 |
| CSF10 | Adoption of procurement system that suits IBS construction method. | 2 | 2 | 33 | 70 | 96 | 0.000 | 1.735 | 0.884 |
| CSF11 | Implementation of standard procedure for onsite and offsite inspection/supervision work. | 0 | 1 | 27 | 73 | 102 | 0.000 | 3.321 | 0.651 |
| CSF12 | Early planning to implement IBS system. | 0 | 4 | 16 | 62 | 121 | 0.000 | 7.461 | 0.189 |
| CSF13 | Team agreement on project deliverables. | 0 | 1 | 27 | 66 | 109 | 0.000 | 2.901 | 0.715 |
| CSF14 | Adoption of standard dimensions and modular coordination to reduce cost. | 1 | 0 | 29 | 60 | 113 | 0.000 | 8.286 | 0.141 |
| CSF15 | Freezing the design early to reduce any possible rework. | 0 | 4 | 31 | 56 | 112 | 0.000 | 1.357 | 0.929 |
| CSF16 | Effective communication and collaboration among players from early phase. | 1 | 2 | 19 | 64 | 117 | 0.000 | 3.259 | 0.660 |
| CSF17 | Project location evaluation and accessibility. | 0 | 0 | 26 | 66 | 111 | 0.000 | 2.854 | 0.723 |
| CSF18 | Increasing the number of high-quality IBS housing units will increase buyers' acceptability. | 1 | 3 | 32 | 75 | 92 | 0.000 | 3.978 | 0.553 |
| CSF19 | Sufficient experience of contractors and designers in IBS. | 1 | 7 | 28 | 68 | 99 | 0.000 | 5.649 | 0.342 |
| CSF20 | Skills and competency of project players. | 1 | 3 | 31 | 66 | 102 | 0.000 | 1.820 | 0.873 |
| CSF21 | Competitive labor wage rate. | 3 | 4 | 32 | 71 | 93 | 0.000 | 1.808 | 0.875 |
| CSF22 | Extended training for local labor to strengthen skills in IBS. | 0 | 0 | 23 | 64 | 116 | 0.000 | 3.851 | 0.571 |
| CSF23 | Ensuring effective design and installation using building information modeling (BIM). | 1 | 0 | 29 | 72 | 101 | 0.000 | 3.601 | 0.608 |
| CSF24 | Continuous R&D to improve current practices and reach competitive advancement. | 0 | 2 | 25 | 67 | 109 | 0.000 | 3.017 | 0.697 |
| CSF25 | Using at least a semi-automated production toward automation and robotic construction. | 2 | 6 | 42 | 75 | 78 | 0.000 | 3.290 | 0.655 |
| CSF26 | Adopting advanced technologies including cloud and real-time collaboration, advanced building materials and internet of things (IoT). | 2 | 3 | 43 | 74 | 81 | 0.000 | 3.083 | 0.687 |

**Table 6.** Critical Success Factors Ranking.

| Code | Critical Success Factor | Mean | Std. Deviation | IRI | Overall Rank |
|---|---|---|---|---|---|
| CSF12 | Early planning to implement IBS system. | 4.468 | 0.746 | 0.896 | 1 |
| CSF22 | Extended training for local labor to strengthen skills in IBS. | 4.448 | 0.711 | 0.892 | 2 |
| CSF16 | Effective communication and collaboration among players from early phase. | 4.429 | 0.783 | 0.89 | 3 |
| CSF17 | Project location evaluation and accessibility. | 4.399 | 0.747 | 0.884 | 4 |
| CSF14 | Adoption of standard dimensions and modular coordination to reduce cost. | 4.379 | 0.802 | 0.88 | 5 |
| CSF13 | Team agreement on project deliverables. | 4.374 | 0.769 | 0.879 | 6 |
| CSF24 | Continuous R&D to improve current practices and reach competitive advancement. | 4.374 | 0.776 | 0.879 | 7 |
| CSF9 | Manufacturer readiness to provide training before installation of components. | 4.36 | 0.76 | 0.874 | 8 |
| CSF11 | Implementation of standard procedure for onsite and offsite inspection/supervision work. | 4.35 | 0.745 | 0.872 | 9 |
| CSF15 | Freeze the design early to reduce any possible rework. | 4.34 | 0.843 | 0.872 | 10 |
| CSF23 | Ensure effective design and installation using building information modeling (BIM). | 4.32 | 0.79 | 0.868 | 11 |
| CSF5 | Provide incentives for IBS implementation including tax privilege and loan support. | 4.291 | 0.873 | 0.862 | 12 |
| CSF20 | Skills and competency of project players. | 4.286 | 0.848 | 0.861 | 13 |
| CSF19 | Sufficient experience of contractors and designers in IBS. | 4.246 | 0.889 | 0.853 | 14 |
| CSF10 | Adoption of procurement system that suits IBS construction method. | 4.251 | 0.851 | 0.852 | 15 |
| CSF18 | Increasing the number of high-quality IBS housing units will increase buyers' acceptability. | 4.232 | 0.839 | 0.85 | 16 |
| CSF3 | Implementation of preferential policy for IBS which can motivate developers and buyers to adopt IBS. | 4.217 | 0.816 | 0.847 | 17 |
| CSF7 | Adopting role and business strategies that support IBS. | 4.202 | 0.767 | 0.844 | 18 |
| CSF21 | Competitive labor wage rate. | 4.197 | 0.912 | 0.843 | 19 |
| CSF1 | Clear plan and policy that can ensure IBS implementation. | 4.163 | 0.831 | 0.836 | 20 |
| CSF8 | Manufacturer involvement role in design and construction. | 4.138 | 0.907 | 0.832 | 21 |
| CSF4 | Adoption of non-financial incentives for IBS, e.g., faster approval procedure, exemption from some building requirements. | 4.119 | 0.876 | 0.828 | 22 |
| CSF26 | Adopting advanced technologies including cloud and real-time collaboration, advanced building materials and internet of things (IoT). | 4.118 | 0.876 | 0.826 | 23 |
| CSF2 | Commitment of agencies and local authorities in the state toward the implementation of IBS policy. | 4.103 | 0.892 | 0.825 | 24 |
| CSF25 | Using at least a semi-automated production toward automation and robotic construction. | 4.079 | 0.903 | 0.818 | 25 |
| CSF6 | Implementing higher taxes and penalties for building waste dumping to reduce dependence on conventional construction. | 3.956 | 1.078 | 0.795 | 26 |

## 5. Discussion

### 5.1. Key Groups of CSFs

A process of mixed analysis through a literature review and interviews produced a list of twenty-six critical success factors grouped into five categories. At first, this research delved into CSFs from an international perspective. Moreover, this analysis was strengthened by focusing on the Malaysian context through the lens of local Malaysian IBS experts, as non-domestic literature could be irrelevant to the situation in Malaysia. Finally, based on semi-structured interview analysis, all twenty-six CSFs of this study were categorized into five groups including (1) planning and control, (2) roles and responsibilities, (3) policies and incentives, (4) industry maturity and (5) technology advancement. The following sections highlight and discuss these components separately.

### 5.1.1. Planning and Control

Six factors under the planning and control group include early planning, team agreement on project deliverables, adoption of standard dimensions, freezing the design early, effective communication and collaboration among players and project location evaluation and accessibility. Motivating construction players to be equipped with IBS skills and capabilities clearly impacts IBS implementation. Additionally, as many low-rise buildings are not in the city center, project location is seen as crucial. All previous factors would facilitate effective communication among project players. To conclude, ensuring a long-term implementation requires early planning and cooperation of all related decision makers to uplift IBS adoption to a higher level [69].

### 5.1.2. Roles and Responsibilities

The second group consists of five factors reflecting the stakeholders' roles and commitments. First, as most companies are profit driven, updating business strategies to suit industrialized development and have a competitive advantage is crucial for survival. Moreover, manufacturers' readiness to provide training is critical since IBS adoption is in the infancy phase; different manufacturers also use their own proprietary systems making a unified installation approach challenging. This is specifically critical since most low-rise buildings are constructed by SMEs who required training prior to commencing the work. Adopting an appropriate procurement system such as design and build will mitigate payment procedure issues. Integrated procurement will enable key stakeholders' engagement, both the contractor and manufacturer from the design phase. In line with that, adopting standard procedures for onsite and offsite work can ensure a smooth process and the manufacturers' critical role in providing IBS components on site safely. However, in the interview, some experts mentioned that most government projects maintained traditional procurement systems and poor payment mechanisms which hindered further development. Dzulkalnine et al. [70] affirmed the need for suitable procurement for IBS projects, as most local contracting companies are small and medium-sized.

### 5.1.3. Policies and Incentives

Successful IBS implementation requires policy intervention and incentives, especially in the industry's early development phase. The third category comprises six factors, and each one reflects different aspects of policies and incentives from the Malaysian IBS construction. Adopting a policy that clearly emphasizes IBS implementation is essential. Yet, more significantly, it is the commitment to that policy. According to the interviews, the federal government does support IBS adoption. Nevertheless, actual implementation relies on local authorities and government agencies that are reluctant toward further IBS applications. Motivating demand-stakeholders such as developers and buyers through preferential policies is crucial. Financial incentives in the form of tax privilege and loan support are presumed to be the most effective method to motivate developers and contractors to adopt industrialized buildings [71]. Low-rise building implementers are in need of support, especially in the early phase of modernizing construction practices. Even

non-financial incentives for IBS projects such as faster approval, reducing the government building requirements and higher plots for IBSs in affordable housing projects can motivate construction stakeholders. Yet, as an added advantage, that will not cause any burden on the government. For further IBS implementation, higher taxes on waste dumping would increase conventional construction costs and lead to sustainable development. In Malaysia, various researchers reported the current weak waste management policies and demanded prudent actions [3,4].

### 5.1.4. Technology Advancement

The fourth category that would enable successful IBS implementation is technology advancement. This group contains four factors that support technological competitiveness toward industrialized construction. First, adopting advanced technologies and building materials would enable Malaysian companies to compete locally and internationally. Using automated and semi-automated manufacturing production will boost productivity. However, interviewees agreed on the need to uplift current manufacturing capabilities but only for long-term planning as that would increase the cost. Additionally, local research and development (R&D) is imminent. Based on the interviews, manufacturers affirm the need for continuous R&D for IBS development. Nevertheless, current low market demand, limited facilities and high financial burdens are making the R&D progress more difficult and slower. This is evident as most low-rise buildings lack the large-scale volume needed to justify R&D investment. Building information modeling (BIM) has the ability to visualize the project leading to better integration of all design teams throughout the design and installation phase. According to Abanda et al. [72], factory-based IBSs would need more integrated information technology procedures and processes.

### 5.1.5. Industry Maturity

Industry maturity is the fifth and final group that can engender successful IBS application. This category integrates five factors measuring the level of IBS maturity in the construction industry. First, smooth IBS development requires incremental improvement to reach a higher level of IBS maturity. Industrialized buildings would become more competitive with the sufficient experience of designers and contractors. Third, project team skills and competency in IBS building would strengthen the industry position. Additionally, skills and competency in IBS design and installation would eliminate any issue of workmanship on site. Fourth, the need for extended training has been considered very critical. The low- and mid-rise building industry is managed by a large number of local companies where the skills and competition need to be nurtured. During the interview, experts mentioned the current limited number of qualified IBS workers due to short, inadequate courses and training provided through government institutions. Furthermore, other experts added that many local workers are given training, but not all trained workers continue to work in the construction industry. Therefore, a competitive labor wage is required to motivate local workers to participate in the building industry. Malaysia's IBS market position is considered relatively immature in comparison to the Sweden and Australian IBS markets [29]. As the industry becomes more mature, the production of high-quality houses becomes cheaper, resulting in stronger demand from buyers and developers [44].

### 5.2. Differences among Respondents' Groups

Investigation of various groups of stakeholders in the IBS industry would provide an inclusive perspective and clarify key differences, if any. In this study, three critical success factors were perceived differently among different building organizations. The first factor is related to the commitment of agencies and local authorities toward IBS implementation. Upon further analysis of the mean of each group, the results indicate that both the government officials and contractors ranked the factor lower compared to manufacturers and consultants. It was found that there is a significant difference between the manufacturers on one side and the government officials and contractors on the other

side. Nonetheless, it was less significant between the consultants and contractors. It must be noted that according to Malaysian government regulations, it is mandatory for all government building projects with a budget over MYR 10 million to implement IBSs [73]. However, the literature indicates that not all public projects adopt IBSs from the Public Work Department (PWD) to other government agencies [26,74]. For government officials, it would be that the government is committed enough toward IBSs, yet it should not blindly adopt IBSs without any concerns for other financial and design issues. In addition, contractors are not specialized in IBSs like manufacturers but take IBSs in addition to conventional construction which is still dominant.

The second factor perceived differently is implementing a preferential policy for IBSs. The manufacturers ranked this factor the highest, while contractors and government officials ranked this factor moderately. As one of the interviewees highlighted, the demand for IBSs in Malaysia is weak without any policy in place to improve that. Compared to certified green buildings, the buyer would receive tax reduction [75], even though IBSs are also regarded as sustainable and green buildings [62,76]. Since manufacturers are working in IBSs mostly, they are the main stakeholders who demand more policies to increase demand, whereby contractors are less concerned with preferential policies as they would work in conventional construction.

Finally, the third factor "implementing higher taxes and penalties for building waste dumping" is perceived differently among different groups of stakeholders and among respondents in general. According to Table 6, CSF6 holds the highest standard deviation that indicates high disagreement among respondents as a whole regardless of their organization type. The factor "implementing higher taxes and penalties for building waste dumping" is considered critical, yet there is disagreement among respondents about its importance. Like previous factors, manufacturers hold a contrary perception to contractors. However, both the government officials and manufacturers ranked the factor similarly. This factor was perceived to hold a negative impact on the general construction work as it would increase the cost burden. Regardless of the potential benefit of IBS adoption and a clean working environment, respondents are in disagreement with pushing for material waste taxes.

### 5.3. Ranking of the CSFs

This section highlights the factors with the highest impact concerning low-mid-rise building projects. Based on the index of relative importance (IRI) of all twenty-six CSFs ranked, the top five factors are (1) early planning to implement IBS system (0.896), (2) extended training for local labor to strengthen skills in IBS (0.892), (3) effective communication and collaboration among players from early phase (0.890), (4) project location evaluation and accessibility (0.884) and (5) adoption of standard dimensions and modular coordination (0.880). Planning, extended training, communication, project location and standard designs are the top factors impacting successful IBS implementation. Nevertheless, according to Tables 5 and 6, all twenty-six CSFs are ranked above IRI = 0.7 affirming that each CSF is considered "highly" critical. Factors with lower ranks do not indicate that they are not important, yet rather highlight the relative importance within the overall CSFs.

### 6. Conclusions

Although industrialized buildings are increasingly recognized as the future to achieve sustainable construction, their application in developing countries such as Malaysia is still lagging. This research investigated the essential factors needed to elevate the IBS success rate and provide clear guidance for the construction stakeholders, especially in the low- and mid-rise building sector.

Consequently, to reach successful implementation, exploratory interviews were initially conducted with several IBS experts to explore the critical success factors (CSFs) in Malaysian residential projects. A few factors were added, while the remaining factors were modified to suit the Malaysian context. Eventually, the interview resulted in twenty-six CSFs categorized into five groups: planning and control, roles and responsibilities, policies

and incentives, technology advancement and industry maturity. These CSFs were assessed by IBS professionals using a survey questionnaire. This study analyzed the differences among construction organizations, identified the factors' relative importance and discussed all groups of CSFs. Based on the survey results, three factors' importance was presumed differently among different organizations including commitment of agencies and local authorities toward IBS policy, implementation of preferential policy for IBS and imposition of higher taxes on waste dumping. This indicates a low understanding of IBS benefits and potential as many Malaysian organizations depend heavily on wasteful conventional construction practices. According to the index of relative importance, the top five factors are: (1) early planning to implement IBS system, (2) extended training for local labor, (3) effective communication among project players, (4) project location evaluation and accessibility and (5) standardized design concept adoption. These factors demonstrate how crucial early planning, skilled labor, communication, project location and standardization are to IBS development.

The study findings provide a comprehensive CSF list based on IBS experts' views and industry professionals, which can help guide the industry stakeholders in IBS planning and execution, especially in developing countries. However, there are some limitations to this study. This research covers the CSFs for IBS stakeholders working in the Malaysian residential sector. Thus, the finding reflects only the local context as the evaluated CSFs are developed based on Malaysian IBS professionals and based on low- and mid-rise residential buildings only. Moreover, future studies need to assess the perspectives on different IBS-based systems such as precast, timber and steel systems.

**Supplementary Materials:** The following supporting information can be downloaded at: https://www.mdpi.com/article/10.3390/su141811711/s1, Table S1: The profile of 15 professionals interviewed.

**Author Contributions:** Conceptualization, A.-H.M.H.A.-A. and N.S.; methodology, A.-H.M.H.A.-A., N.S. and Y.Y.A.-A.;formal analysis, A.-H.M.H.A.-A. and A.-B.A.A.-M.; investigation, A.-H.M.H.A.-A. and A.O.B.; writing—original draft preparation, A.-H.M.H.A.-A.; writing—review and editing, Y.Y.A.-A., A.-B.A.A.-M.; supervision, N.S.; project administration, A.O.B.; funding acquisition, N.S. All authors have read and agreed to the published version of the manuscript.

**Funding:** This research received no external funding.

**Institutional Review Board Statement:** Not applicable.

**Informed Consent Statement:** Not applicable.

**Data Availability Statement:** Not applicable.

**Acknowledgments:** The authors would like to appreciate the Centre for Graduate Studies, Universiti Teknologi PETRONAS (UTP) and International Collaborative Research Fund, UTP of fund number 015ME0–248, for the support provided for this paper.

**Conflicts of Interest:** The authors declare no conflict of interest.

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
