# Peer review of "Essential Factors Enhancing Industrialized Building Implementation in Malaysian Residential Projects"

_sustainability, doi:10.3390/su141811711_

Round 1

Reviewer 1 Report

Considering the global housing crisis, the study could be relevant since it focuses on industrialised building development in housing. The manuscript is well written and well structured. That said, I have some comments for the consideration of the authors for improving the quality of the manuscript.

1. Concerning the introduction, the authors are requested to briefly inform potential readers on the practical and theoretical contributions of the study before informing them on the structure of the paper.

2. The first paragraph of the conclusion reads like the paragraph for an introduction. This could be deleted or improved. Unlike an introduction, the conclusion mostly starts by stating the aim that was achieved in the study.

3. The authors are encouraged to conduct another thorough proofread of the manuscript to correct some minor grammatical errors.  

Author Response

Kindly see the attached file.

Reviewer 2 Report

See the attached document.

Author Response

Kindly see the attached file

Reviewer 3 Report

Presented article is valuable to scientific community. However, there are some elements of the article which require improvements. They are described further.

Even though I don’t feel qualified to judge about English, there are places where the language is sloppy and contain some mistakes, for instance, lines 56, 62, 87, 219, 223. Table 1 - remove “.” at the end of “Purpose of Use” column. Formatting of Fig. 1 needs to be adjusted to a more applicable framework – font and font colour needs to comply with the text of the document. It cannot be that one designation is in black colour, the rest – in white. Formatting of Table 2 should be improved. Critical success factors (Table 4) need to be formulated as factors, not activities to be done (“Implementation of…”, not “implement”; “adoption of”, not “adopt”; and so on). These changes need to be adjusted also further in the manuscript where critical success factors are tackled. Formatting of Figure 2 needs to be improved. Table 4 needs to fit in 1 page. Probably it could be considered for Appendixes. Figure 3 is hardly understandable for readers. It would be better to see the “Zoom in” version covering values from 0,6-1 on Y axis and 4-5 on X axis.

Author Response

Kindly find the attached file.

Reviewer 4 Report

Title: I'm having a little trouble understanding the meaning of the title. It needs to be rewritten in my opinion.

L81-84: Perhaps, it will be useful for the reader to have some precisions already at this level concerning the methodology used and the reasons having led to the choice of the methodology.

L102 Give the meaning of the acronym SOHO

Table 1, Table 2 and related text : To complete data, it will be interesting to have  surfaces of each type of property.

The Table 3 can be put as supplementary material or information.

L196-197. Concerning statistical methods used can you give more details in the part methods and move L233-L254 in this part for example.

Figure 2: This figure is not essential because you give the values in the text. it can be deleted.

Equations 1 and 2 are well known and can be deleted.

Equation 3: define A.

L282 The lowest value for IRI according to equation 3 is 1/5 when all collected responses assigned 1 and the highest is 1 when all responses are 5 but I don't understand how to get zero.

Figure 3 is not interesting in my opinion because all the values are already in table 6 and the values are very close.

L322 -333: Only the first and last sentences are interesting in this paragraph and suffice in my opinion for discussion.

In general, try to avoid overly repetitive sentences.

Paragraph 5.3: The ranking is not significant and therefore of little interest.

Author Response

Kindly find the attached file.

Reviewer 5 Report

Review for sustainability-1882669

It is a valuable study that can fill the gap and provide decision-makers with more perspectives. However, several minor issues should be addressed before it can be accepted:

1. In Line 15, Line 74, and Line 79, the authors mention that "very few have focused on low & mid-rise residential buildings. Is it convincing? In addition, the paper should compare critical success factors applications and industrialized building development of low & mid-rise residential buildings with other buildings. The characters of low & mid-rise residential buildings should be presented

2. In the Method Section, a detailed workflow is suggested.

3. In Table 4. some factors are from the literature review. However, it is highly suggested to present which factors are both mentioned in the literature review and interview. How many times a factor is mentioned in the interview and literature review should be presented.

4. In Table 4 and the discussion section, five categories are important in this article. How did you come up with these categories? Is it through your own summaries and experiences, literature reviews, or interviews? Did participants confirm these categories?

5. The discussion section should emphasize the characteristics of the low & mid-rise residential buildings. It seems that some discussions and suggestions can also be used for other types of buildings.

6. The discussions and conclusions should not be limited to every single category. Some cross-category discussions and findings should be presented. For example, could policies and incentives stimulate technological advancement in the field of CSF, such as BIM and IoT?

Author Response

Kindly see the attached file.

Round 2

Reviewer 4 Report

This paper can be published